# Equivariant Single View Pose Prediction Via Induced and Restricted Representations

Owen Howell [*1], David Klee[2], Ondrej Biza[2], Linfeng Zhao[2], and Robin Walters[2]

[1] Department of Electrical and Computer Engineering, Northeastern University
[2] Khoury College of Computer Sciences, Northeastern University

## Abstract

Learning about the three-dimensional world from two-dimensional images is a fundamental problem in computer vision. An ideal neural network architecture for such tasks would leverage the fact that objects can be rotated and translated in three-dimensions to make predictions about novel images. However, imposing $SO(3)$-equivariance on two-dimensional inputs is difficult because the group of three-dimensional rotations does not have a natural action on the two-dimensional plane. Specifically, it is possible that an element of $SO(3)$ will rotate an image out of the plane. We show that an algorithm that learns a three-dimensional representation of the world from two-dimensional images must satisfy certain consistency properties which we formulate as $SO(2)$-equivariance constraints. We use the induced and restricted representations of $SO(2)$ on $SO(3)$ to construct and classify architectures that satisfy these consistency constraints. We prove our construction realizes all possible architectures that respect these constraints. We show that three previously proposed neural architectures for 3D pose prediction are special cases of our construction. We propose a new model that generalizes previously considered methods and contains additional trainable parameters. We test our architecture on three pose prediction tasks and achieve SOTA results on both the PASCAL3D+ and SYMSOL pose estimation tasks.

## 1  Introduction

One of the fundamental problems in computer vision is learning representations of 3D objects from 2D images [1–3]. By understanding how image features correspond to a physical object, a model can generalize better to novel views of the object, for instance, when estimating the pose of an object. In general, neural networks that respect the symmetries of a problem are more noise robust and data efficient, while also less prone to over-fitting [4]. Three-dimensional space has a natural symmetry group of three-dimensional rotations and three-dimensional translations, $SE(3)$. While we would like to leverage this symmetry to design improved neural architectures, serious challenges exist in incorporating 3D symmetry when applied to image data. Specifically, a projection of a three-dimensional scene into a two-dimensional plane does not transform equivariantly under all elements of $SE(3)$. This is because there is no a priori model for how two-dimensional images transform under out-of-plane object rotations. Cohen and Welling [5] showed how to design neural networks that are explicitly $SO(2)$-equivariant and accept images as inputs. However, $SO(2)$-equivariant methods ignore the fact that the group $SO(3)$ acts on the space of pose configurations.

Equivariant neural networks are much more constrained than general multi-layer perceptrons. The requirement of equivariance to a group $G$ places strict restrictions on the allowed linear maps and the

---

*howell.o@northeastern.edu

37th Conference on Neural Information Processing Systems (NeurIPS 2023).

allowed non-linear functions in each network layer [5, 6]. Because of this, the possible structures of $G$-equivariant neural networks can be completely classified based on the representation theory of the group $G$ [4, 7, 8]. For compact groups, it is possible to characterize the structure of all possible kernels of $G$-equivariant networks [8].

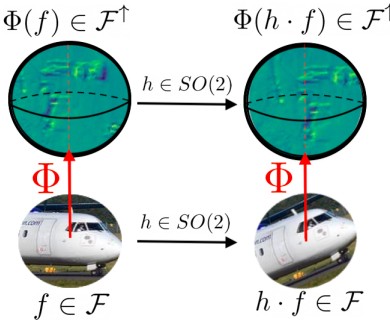

Figure 1: A map $\Phi : \mathcal{F} \to \mathcal{F}^{\uparrow}$ from signals on $\mathbb{R}^2$ to signals on $S^2$. Let $SO(2)$ be the subgroup that consists of all in-plane rotations ( i.e. about the axis defined by the red arrow). The map $\Phi$ must be equivariant with respect to this $SO(2) \subseteq SO(3)$ subgroup.

We argue that any equivariant machine learning algorithm that builds a three-dimensional model of the world from two-dimensional images must satisfy a natural geometric consistency property. This consistency property can be stated as $SO(2)$-equivariance with respect to only the $SO(2)$ subgroup acting along the camera viewing orientation (see Figure 1). We give a complete characterization of maps that satisfy this property. This derivation uses the so-called restricted representation of $SO(2)$ since the group action is restricted from the full $SO(3)$. Using the Frobenius Reciprocity theorem, we show that this geometric constraint can also be derived using induced representations, a type of representation of $SO(3)$ constructed from representations of $SO(2)$. The classification theorems derived in [5, 8, 4] assume that both the input and output layers are $G$-equivariant. The construction presented in [4] is different; we map $H$-equivariant functions to $G$-equivariant functions. Our arguments using induced and restricted representations give a natural generalization of equivariant maps between different groups. We derive analogies of the theorems presented in [9, 8] for the induced and restricted representations.

## 1.1 Importance and Contribution

In this work, we will show how induced and restricted representations can be used to construct neural architectures that accept image data and leverage $SO(3)$-equivariant methods to avoid learning nuisance transformations in three-dimensional space.

We show that our construction satisfies two desirable theoretical properties, *completeness* and *universality*. Let $H \subseteq G$. We focus on the case $G = SO(3)$ and let $H = SO(2)$ but we give a theoretical analysis for general groups. Specifically, the induced representation construction is *complete* in that all group-valued functions on $G$ can be induced from a set of group-valued functions on $H$. The construction is *universal* in that all multi-linear maps that map $H$-equivariant functions to $G$-equivariant functions are specific cases of the induced representation, modulo isomorphism. Furthermore, we show that the architectures proposed in [10, 11] are special cases of our construction for the icosahedral group $G = A_5$ and the construction proposed in [12] is a special case of our construction for the three-dimensional rotation group $G = SO(3)$. Our method achieves state of the art performance for orientation prediction on PASCAL3D+ [13] and SYMSOL [14] datasets.

**Contributions:**

- We propose a unified theory for learning three-dimensional representations from two-dimensional images. We show that algorithms that learn three-dimensional representations from two-dimensional images must satisfy certain consistency properties, which are equivalent to $SO(2)$-steerability constraints.

- We introduce a fully differentiable layer called an *induction/restriction layer* that maps signals on the plane into signals on the sphere. We show that the induction/restriction layer satisfies a natural consistency constraint and prove both a completeness and universal property for our construction.

- Our method achieves SOTA performance for orientation prediction on PASCAL3D+ and SYMSOL datasets.

## 2 Related Work

**Equivariant Learning** Incorporating task symmetry into the design of neural networks has been effective in domains such as computer vision [15, 16], point cloud processing [17, 18], and robotics [19]. Cohen and Welling [9] introduced steerable kernels which are a trainable layer that can be used to build networks that are equivariant to 2D [5, 20] and 3D transformations [21, 22]. The majority of past works have studied end-to-end equivariant models, where the input can be transformed by the action of the group and all layers are equivariant, in this work, we explore how to 'fuse' $SO(2)$ and $SO(3)$ equivariant layers.

There has been growing interest in leveraging 3D symmetry from 2D inputs. [23, 24] learned a 3D transformable latent space from images of a single object. [25] trained a convolutional network to predict pre-trained $SO(3)$ equivariant embeddings, while [11, 10, 12] mapped image features onto elements of the discrete group of $SO(3)$, using structured viewpoints or a hand-coded projections, respectively. In contrast to prior work, we provide a theoretical foundation for learned equivariant mappings from 2D to 3D, which additionally guides us to introduce a more general and effective trainable operation.

**Object Pose Estimation** Predicting the 3D orientation of objects is an important problem in fields like autonomous driving [26], robotics [27] and cryogenic electron microscopy [28]. Many works [29, 30] have used a regression approach, and others [31–33] have identified ways to mitigate the discontinuities along the $SO(3)$ manifold. More recent works have explored ways to model orientation as a distribution over 3D rotations, which handles object symmetries and captures uncertainty. [34], [35] and [36] predict parameters for Bingham, von Mises and Laplace distributions, respectively. These families of distributions can have limited expressivity, so other work explored using implicit networks [14] or the Fourier basis [12] to model more complex pose distributions. Inspired by [12, 23], we parameterize the latent object pose as a distribution on $SO(3)$ and then ask what constraints need to be imposed to enforce $SO(2) \subseteq SO(3)$-equivariance A.0.4, which is a generalization of $SO(2)$-equivariance.

## 3 Background

We introduce the induced and restricted representations. For a more extensive review of representation theory, see A.

Let $G$ be a group and $V$ be a vector space over $\mathbb{C}$. A *representation* $(\rho, V)$ of $G$ is a map $\rho : G \to \mathrm{Hom}[V, V]$ such that

$$\forall g, g' \in G, \ \ \forall v \in V, \quad \rho(g \cdot g')v = \rho(g) \cdot \rho(g')v$$

Concisely, a group representation is an embedding of a group into a set of matrices. The matrix embedding must obey the multiplication rule of the group. We now introduce the *restricted representation* and *induced representation*.

**Restricted Representation** Let $H \subseteq G$ be a subgroup. Let $(\rho, V)$ be a representation of $G$. The restricted representation of $(\rho, V)$ from $G$ to $H$ is denoted as $\mathrm{Res}_H^G[(\rho, V)]$. Intuitively, $\mathrm{Res}_H^G[(\rho, V)]$ can be viewed as $(\rho, V)$ evaluated on the subgroup $H$ of $G$. Specifically,

$$\forall h \in H, \ \ \forall v \in V, \quad \mathrm{Res}_H^G[\rho](h)v = \rho(h)v$$

For a more in depth discussion of the restricted representation, please see A.

**Induced Representation** The induced representation is a way to construct representations of a larger group $G$ out of representations of a subgroup $H \subseteq G$. Let $(\rho, V)$ be a representation of $H$. The induced representation of $(\rho, V)$ from $H$ to $G$ is denoted as $\mathrm{Ind}_H^G[(\rho, V)]$. Define the space of functions

$$\mathcal{F} = \{ \ f \ | \ f : G \to V, \ \forall h \in H, \ \ f(gh) = \rho(h^{-1})f(g) \ \}$$

Then the induced representation is defined as $(\pi, \mathcal{F}) = \mathrm{Ind}_H^G[(\rho, V)]$ where the induced action $\pi$ acts on the function space $\mathcal{F}$ via

$$\forall g, g' \in G, \ \ \forall f \in \mathcal{F}, \quad (\pi(g) \cdot f)(g') = f(g^{-1}g')$$

Please see A for an in depth discussion of the induced representation. The induced and restricted representations are adjoint functors [37].

## 4 Method

Convolutional networks or vision transformers are typically used to extract spatial feature maps from 2D images. For convenience we ignore discretization and treat the feature maps as having continuous inputs $f : \mathbb{R}^2 \to \mathbb{R}^d$. To leverage spatial symmetries in 3D, we would like to map our features $f$ from a plane onto a sphere: $g : S^2 \to \mathbb{R}^D$. Klee et al. [12] proposed one such mapping, where the planar feature map is stretched over a hemisphere, but other possible mappings exist.

We formalize the equivariance property that every projection should have through the theory of induced and restricted representations. The constraints that we impose have an intuitive geometric interpretation. We give a complete characterization of *all possible* linear and equivariant projections $\Phi$ from planar features to a spherical representation. Our general formulation includes [12] as a special case, and we show that a learnable equivariant projection leads to better predictive models.

### 4.1 Equivariant 2D to 3D Projection by Induced and Restricted Representations

We first derive the $SO(2)$-equivariance constraint for the most general linear mapping from images to spherical signals.

**Image inputs**    We first describe $\mathcal{F}$ the space of image input signals. Let $V$ and $V^\uparrow$ be vector spaces. Let $\mathcal{F}$ be the vector space of all $V$-valued signals defined on the plane

$$\mathcal{F} = \{ \ f \ | \ f : \mathbb{R}^2 \to V \ \}.$$

Elements of $\mathcal{F}$ are sometimes called $SE(2)$-steerable feature fields [20]. The group $SE(2) = \mathbb{R}^2 \rtimes SO(2)$ of 2D translations and rotations acts on $\mathcal{F}$ via representation $\pi$. Each $h \in SE(2)$ has a unique factorization $h = \bar{h} h_c$ where $\bar{h} \in \mathbb{R}^2$ is a translation and $h_c \in SO(2)$ is a rotation. Let $(\rho, V)$ be an $SO(2)$-representation describing the transformation of the fibers of the features $f$. Then the action $\pi$ is defined

$$\forall f \in \mathcal{F}, \ r \in \mathbb{R}^2, \ h \in SE(2), \quad \pi(h) \cdot f(r) = \rho(h_c) f(h^{-1} r)$$

so that $(\pi, \mathcal{F}) = \mathrm{Ind}_{SO(2)}^{SE(2)} [(\rho, V)]$ and $(\pi, \mathcal{F})$ gives a representation of the group $SE(2)$ [9].

**Spherical outputs**    We would like to map signals in $\mathcal{F}$ to functions from $S^2$ into the vector space $V^\uparrow$. Let $\mathcal{F}^\uparrow$ be the vector space of all such outputs defined as

$$\mathcal{F}^\uparrow = \{ \ f \ | \ f : S^2 \to V^\uparrow \ \}$$

The group $SO(3)$ acts on the vector space $\mathcal{F}^\uparrow$ via

$$\forall f^\uparrow \in \mathcal{F}^\uparrow, \ \hat{n} \in S^2, \ g \in SO(3), \quad \pi^\uparrow(g) \cdot f^\uparrow(\hat{n}) = \rho^\uparrow(g) f^\uparrow(g^{-1} \hat{n})$$

where $\rho^\uparrow(g)$ describes the $SO(3)$ fiber representation.

$SO(2)$**-equivariant image to sphere**    Let $H = SO(2)$ be the $SO(2)$ subgroup of $SO(3)$ that corresponds to in-plane rotations of the image. Our goal is to classify $H$-equivariant linear maps $\Phi : \mathcal{F} \to \mathcal{F}^\uparrow$. This is equivalent to the constraint that

$$\forall h \in H = SO(2), \ f \in \mathcal{F}, \quad \Phi(\pi(h) \cdot f) = \pi^\uparrow(h) \cdot \Phi(f) \tag{1}$$

The constraint enforces equivariance with respect to $SO(2)$ transformations. By definition, the evaluation of $\pi^\uparrow(h)$ at $h \in SO(2)$ is the restricted representation $\pi^\uparrow(h) = \mathrm{Res}_{SO(2)}^{SO(3)} [\pi^\uparrow](h)$.

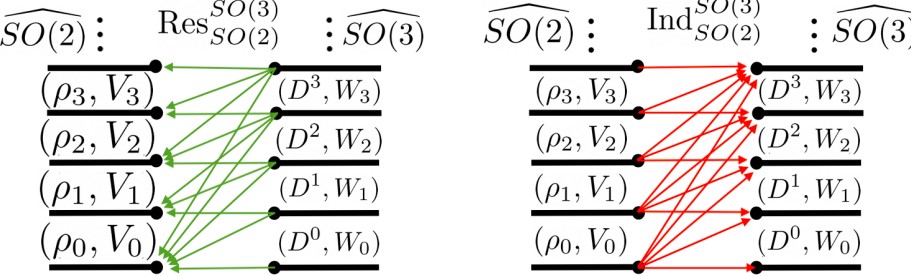

**Figure 2: Left**: Decomposition of the restricted representation $\text{Res}_{SO(2)}^{SO(3)}$ of SO(3)-irreducibles $(D^\ell, W_\ell) \in \widehat{SO(3)}$ into SO(2)-irreducibles $(\rho_k, V_k) \in \widehat{SO(2)}$. Not every SO(2)-representation can be realized as the restriction of a SO(3)-representation. **Right**: Decomposition of the induced representation $\text{Ind}_{SO(2)}^{SO(3)}$ for SO(2)-irreducibles $(\rho_k, V_k) \in \widehat{SO(2)}$ into SO(3)-irreducibles $(D^\ell, W_\ell) \in \widehat{SO(3)}$. Not every SO(3)-representation can be realized as the induction of an SO(2)-representation.

## 4.2 Solving the Kernel Constraint

We use tools from [38, 8] to solve for the space of all possible maps satisfying the constraint 1, giving the trainable space for the image to sphere layer.

Our conclusion is that instead of mapping arbitrary SO(2)-input representations to arbitrary SO(2)-output representations, the allowed input and output representations $(\rho, V)$ and $(\rho^\uparrow, V^\uparrow)$ must satisfy additional constraints. Specifically, not every representation can be realized as the restriction of an SO(3) to SO(2) representation 2. Although in this paper we focus on orientation estimation, the equivariant framework in Section C.0.1 is more general. In the Appendix D, we formulate and solve analogous equivariance constraints for both 6DoF-pose estimation and monocular volume reconstruction.

**Theorem 1.** *The constraint in Equation 1 can be solved exactly using the results of [38, 8]. The most linear general map $\Phi : \mathcal{F} \to \mathcal{F}^\uparrow$ can be expanded as*

$$[\Phi(f)](\hat{n}) = \int_{r \in \mathbb{R}^2} dr\ \kappa(\hat{n}, r) f(r)$$

*where $\kappa : \mathbb{R}^2 \times S^2 \to \text{Hom}[V, V^\uparrow]$. Then, the exact form of $\kappa$ can be written as*

$$\kappa(\hat{n}, r) = \sum_{\ell=0}^{\infty} F_\ell(r)^T Y_\ell(\hat{n}) \tag{2}$$

*where $Y_\ell(\hat{n})$ is the vectorization of the $\ell$-type spherical harmonics and each $F_\ell(r)$ is a standard SO(2)-steerable kernel [9, 38] that has input SO(2)-representation $(\rho, V)$ and output SO(2)-representation $(\rho^\ell, V^\ell) = (\rho, V) \otimes \text{Res}_{SO(2)}^{SO(3)}[(D^\ell, V^\ell)]$.*

The proof of this statement is given in Appendix F. Note that similar to [18, 6] the tensor product structure of the SO(2) and SO(3) irreducible representations determine the allowed input and output representations of the matrix valued harmonic coefficients $F_\ell(r)$.

## 4.3 Including Non-Linearities

In Section 4.2, we considered the most general linear maps that satisfied the generalized equivariance constraint. Adding non-linearities should allow for more expressiveness. Understanding non-linearities between equivariant layers is still an active area of research [39–42].

One way to include non-linearity is to apply standard SO(3) non-linearities after the linear induction layer. After applying the linear mapping described in C, we apply an additional spherical non-linearity [43] to the signal on $S^2$. This is the method we employ for the results presented in 6.2. As shown in G it is also possible to include tensor-product based non-linearity analogous to the results of [18, 6].

## 5 Theory

### 5.1 Universal Property

In section 4 we showed how the restriction representation arises naturally when constructing $\mathrm{SO}(3)$-equivariant architectures for image data. However, there is no a priori choice of the hidden $\mathrm{SO}(3)$ representation. We show that with this choice, our construction satisfies a universal property and is unique up to isomorphism [44].

A standard result in group theory establishes the following universal property of induced representations, as stated in [37]:

**Theorem 2.** *Let $H \subseteq G$. Let $(\rho, V)$ be any $H$-representation. Let $Ind_H^G(\rho, V)$ be the induced representation of $(\rho, V)$ from $H$ to $G$. Then, there exists a unique $H$-equivariant linear map $\Phi_\rho : V \to Ind_H^G V$ such that for any $G$-representation $(\sigma, W)$ and any $H$-equivariant linear map $\Psi : V \to W$, there is a unique $G$-equivariant map $\Psi^\uparrow : Ind_H^G V \to W$ such that the diagram 3 is commutative.*

This theorem can be applied to the construction proposed in 1 to prove a universality property, similar to the results of [5] for $G$-equivariant neural networks.

$$(\rho, V) \xrightarrow{\ \Phi_\rho\ } \mathrm{Ind}_H^G(\rho, V)$$
$$\Psi \searrow \quad \downarrow \Psi^\uparrow$$
$$(\sigma, W)$$

Figure 3: Commutative Diagram for Uniqueness Property of Induced Representations.

Let $(\rho, V)$ be a $H$-representation and let $(\sigma, W)$ be a $G$-representation. Let $\Psi : V \to W$ where $\Psi$ is an intertwiner of a the $H$-representation and the restriction of the $G$-representation to an $H$-representation so that

$$\forall h \in H, \quad \Psi \rho(h) = \mathrm{Res}_H^G[\sigma](h)\Psi$$

so that $\Psi \in \mathrm{Hom}_H[(\rho, V), \mathrm{Res}_H^G(\sigma, W)]$. The universal property of the induced representation allows us to write any such $\Psi$ in a canonical form. Specifically, as illustrated in Figure 5.1, we can always uniquely decompose $\Psi = \Psi^\uparrow \circ \Phi_\rho$ where $\Psi^\uparrow \in$ $\mathrm{Hom}_G[\mathrm{Ind}_H^G(\rho, V), (\sigma, W)]$ and $\Psi_\rho : V \to \mathrm{Ind}_H^G V$ is $(\sigma, W)$ independent.

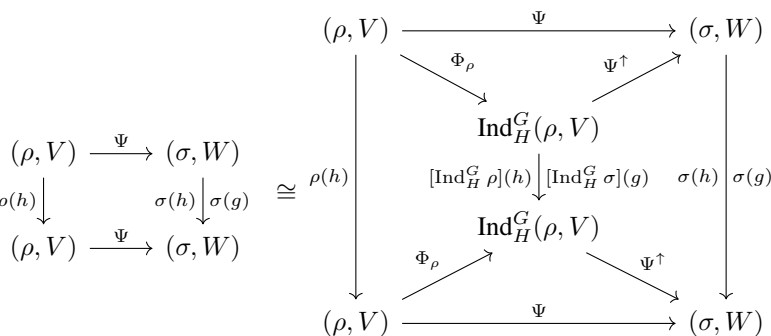

Figure 4: Factorization Identity for Universal Property of Induced Representations

**Factorization Property of $H \subseteq G$ Neural Networks**   We use the universal property of induced representations to show that all possible latent $G$-equivariant architectures can be written in terms of the induced representation. At each layer of a equivariant neural network, we have a set of functions from a homogeneous space of a group into some vector space [6]. Let $X_i^H$ be a set of homogeneous spaces of the group $H$ and let $X_j^G$ be a set of homogeneous spaces of the group $G$. Let $V_i^H$ and $W_j^G$ be a set of vector spaces. Then, consider the function spaces

$$\mathcal{F}_i^H = \{\ f\ |\ f : X_i^H \to V_i^H\ \}, \qquad \mathcal{F}_j^G = \{\ f'\ |\ f' : X_j^G \to W_j^G\ \}$$

The group $H$ acts on the homogeneous spaces $X_i^H$ and the group $G$ acts on the homogeneous spaces $X_j^G$ so that the function spaces $\mathcal{F}_i^H$ and $\mathcal{F}_j^G$ form representations of $H$ and $G$, respectively

Suppose we wish to design a downstream $G$-equivariant neural network that accepts as signals functions that live in the vector space $\mathcal{F}_0^H$ and transform in the $\rho_0$ representation of $H$. Thus,

$(\rho_0, \mathcal{F}_0^H)$ is a $H$-representation, but not necessarily a $G$-representation. At some point, in the architecture, a layer $\mathcal{F}_i^H$ must be $H$ equivariant on the left and both $H$ and $G$-equivariant on the right. Let us call the layer that is both $H$ and $G$-equivariant $\mathcal{F}_1^G$.

$$\ldots \xrightarrow{\Phi_{i-1}} (\rho_i, \mathcal{F}_i^H) \xrightarrow{\Psi} (\sigma_1, \mathcal{F}_1^G) \xrightarrow{\Psi_1} \ldots \qquad \ldots \xrightarrow{\Phi_{i-1}} (\rho_i, \mathcal{F}_i^H) \xrightarrow{\Phi_{\rho_i}} \mathrm{Ind}_H^G[(\rho_i, \mathcal{F}_i^H)] \xrightarrow{\Psi^\uparrow} (\sigma_1, \mathcal{F}_1^G) \xrightarrow{\Psi_1} \ldots$$

$$\downarrow{\rho_i(h)} \quad \downarrow{\sigma_1(g)} \qquad \cong \qquad \downarrow{\rho_i(h)} \quad \downarrow{\mathrm{Ind}_H^G[\rho_i]} \quad \downarrow{\sigma_1(g)}$$

$$\ldots \xrightarrow{\Phi_{i-1}} (\rho_i, \mathcal{F}_i^H) \xrightarrow{\Psi} (\sigma_1, \mathcal{F}_1^G) \xrightarrow{\Psi_1} \ldots \qquad \ldots \xrightarrow{\Phi_{i-1}} (\rho_i, \mathcal{F}_i^H) \xrightarrow{\Phi_{\rho_i}} \mathrm{Ind}_H^G[(\rho_i, \mathcal{F}_i^H)] \xrightarrow{\Psi^\uparrow} (\sigma_1, \mathcal{F}_1^G) \xrightarrow{\Psi_1} \ldots$$

Figure 5: Factorization of Generic Architecture Using Universal Property of Induced Representation. Any network that has input layer $(\rho_i, \mathcal{F}_i^H)$ that is $H$-equivariant and output layer $(\sigma_1^G, \mathcal{F}_1^G)$ that is $G$-equivariant can be factorized in terms of the induced representation. The map $\Psi = \Psi^\uparrow \circ \Phi_{\sigma_i}$ where $\Psi^\uparrow$ is $G$-equivariant and $\Phi_{\sigma_i}$ is $H$-equivariant.

Suppose that $\Psi$ is an intertwiner between $(\rho_i, \mathcal{F}_i^H)$ and $(\sigma_1, \mathcal{F}_1^G)$. Using the factorization property of induced representations 5.1, there is a canonical basis of the space $\mathrm{Hom}_H[(\rho_i, \mathcal{F}_i^H), \mathrm{Res}_H^G[(\sigma_1, \mathcal{F}_1^G)]] \cong \mathrm{Hom}_G[\mathrm{Ind}_H^G[(\rho_i, \mathcal{F}_i^H)], (\sigma_1, \mathcal{F}_1^G)]$ and we may write $\Psi$ uniquely as $\Psi = \Psi^\uparrow \circ \Phi_\rho$ where $\Phi_\rho$ is an $H$-equivariant map and $\Psi^\uparrow$ is a $G$-equivariant map. Thus, any boundary between $H$ and $G$ layers can be written as an $H$-equivariant layer between $(\rho_i, \mathcal{F}_i^H)$ and $\mathrm{Ind}_H^G[(\rho_i, \mathcal{F}_i^H)]$ followed by a $G$-equivariant layer between $\mathrm{Ind}_H^G[(\rho_i, \mathcal{F}_i^H)]$ and $(\sigma_1, \mathcal{F}_1^H)$. In this way, induction is all you need and all possible latent $G$-equivariant architectures can be written in terms of the induction representation.

# 6 Experiments

## 6.1 Datasets & Evaluation Metrics

We evaluate the performance of our method on three single-object pose estimation datasets. These datasets require making predictions in $\mathrm{SO}(3)$ from single 2D images. **SYMSOL** [14] consists of a set of images of marked and unmarked geometric solids, taken from different vantage points. Training data is annotated with viewing direction. Some objects have symmetries so that there are multiple equivalent viewing directions, which requires learning distributions over poses. **PASCAL3D+** [13] is a popular benchmark for object pose estimation composed of real images of objects from twelve categories. This dataset is challenging due to the large variation in object appearances and the presence of novel object instances in the test set. To be consistent with the baselines, we augment the training data with synthetic renderings[45] and evaluate performance on the PASCALVOC_val set. For more details on the benchmark datasets and additional numerical experiments, see B.

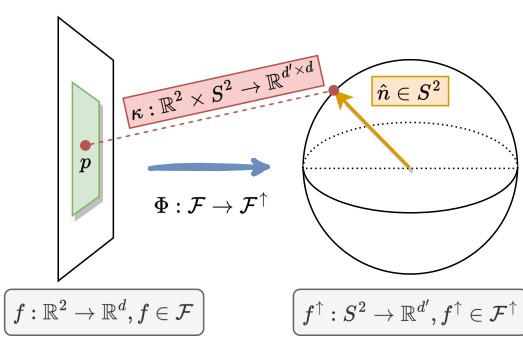

$\kappa : \mathbb{R}^2 \times S^2 \to \mathbb{R}^{d' \times d}$

$\hat{n} \in S^2$

$p$

$\Phi : \mathcal{F} \to \mathcal{F}^\uparrow$

$f : \mathbb{R}^2 \to \mathbb{R}^d, f \in \mathcal{F}$ $\qquad$ $f^\uparrow : S^2 \to \mathbb{R}^{d'}, f^\uparrow \in \mathcal{F}^\uparrow$

Figure 6: Diagram of an Equivariant Image to Sphere Convolution. At each unit vector $\hat{n} \in S^2$ the kernel $\kappa(\hat{n} : p)$ is dependent on the image point $p = (x, y) \in \mathbb{R}^2$. Equivariance constraints put restrictions on the allowed form of $\kappa(\hat{n} : p)$. Similar to a standard convolution, the kernel $\kappa$ has a user defined receptive field.

When a single ground truth rotation label is provided, we evaluate the method using the geodesic distance between the predicted and ground truth rotation matrices, reported as either median rotation error or accuracy at a given rotation error threshold. For SYMSOL, which provides the full set of equivalent rotations associated with an image, we measure the accuracy of the learned pose distribution using average log-likelihood. This is also the accuracy metric used in [12].

## 6.2 Implementation & Training Details

For the results presented in 6, we use a ResNet encoder with weights pre-trained on ImageNet. With 224x224 images as input, this generates a 7x7 feature map with 2048 channels.

The filters in the induction layer are instantiated using the e2nn [38] package. The maximum

frequency is set at $\ell = 6$. The output of the induction layer is a 64-channeled $S^2$ signal with fibers transforming in the trivial representation of SO(3). After the induction layer, a spherical convolution operation is performed using a filter that is parameterized in the Fourier domain, which generates an 8-channel signal over SO(3). A spherical non-linearity is applied by mapping the signal to the spatial domain, applying a ReLU, then mapping back to the Fourier domain. One final spherical convolution with a locally supported filter is performed to generate a one-dimensional signal on SO(3). The output signal is queried using an SO(3) HEALPix grid (recursion level 3 during training, 5 during evaluation) and then normalized using a softmax following [14]. $S^2$ and SO(3) convolutions were performed using the e3nn [43] package. The network was initialized and trained using PyTorch [46].

In order to create a fair comparison to existing baselines, batch size (64), number of epochs (40), optimizer (SGD) and learning rate schedule (StepLR) were chosen to be the same as that of [12]. Numerical experiments were implemented on NVIDIA P-100 GPUs.

### 6.3 Comparison to Baselines

We compare our method's performance to competitive pose estimation baselines. We include regression methods, [29, 30, 33], that perform well on datasets where objects have a single valid pose (e.g. are non-symmetric or symmetry is disambiguated in labels). We also baseline against methods that model pose with parametric families of distributions, [35, 47, 34, 36], an implicit model [14], and the Fourier basis of $SO(3)$ [12]. To make the comparison fair, all methods use the same-sized ResNet backbone for each experiment, and we report results as stated in the original papers where possible.

**SYMSOL Results** Performance on the SYMSOL dataset is reported in Table 1. Our method achieves the highest average log-likelihood on SYMSOL I. Importantly, we observe a significant improvement over Klee et al. [12] on all objects, which indicates that our induction layer is more effective than its hand-designed orthographic projection. On SYMSOL II, our method slightly underperforms Murphy et al. [14], which has much higher expressivity on the output since it is an implicit model. However, we demonstrate that our approach, which preserves the symmetry present in the images, is better with less data, as shown in Table 2.

Table 1: Average log likelihood (the higher the better ↑) on SYMSOL I & II. Per [14], a single model is trained on all classes in SYMSOL I and a separate model is trained on each class in SYMSOL II.

| | SYMSOL I (↑) | | | | | SYMSOL II (↑) | | | |
| --- | --- | --- | --- | --- | --- | --- | --- | --- | --- |
| | *avg* | *cone* | *cyl* | *tet* | *cube* | *ico* | *avg* | *sphX* | *cylO* | *tetX* |
| Deng et al. [34] | -1.48 | 0.16 | -0.95 | 0.27 | -4.44 | -2.45 | 2.57 | 1.12 | 2.99 | 3.61 |
| Prokudin et al. [35] | -1.87 | -3.34 | -1.28 | -1.86 | -0.50 | -2.39 | 0.48 | -4.19 | 4.16 | 1.48 |
| Gilitschenski et al. [48] | -0.43 | 3.84 | 0.88 | -2.29 | -2.29 | -2.29 | 3.70 | 3.32 | 4.88 | 2.90 |
| Murphy et al. [14] | 4.10 | 4.45 | **4.26** | 5.70 | 4.81 | 1.28 | **7.57** | **7.30** | **6.91** | **8.49** |
| Klee et al. [12] | 3.41 | 3.75 | 3.10 | 4.78 | 3.27 | 2.15 | 4.84 | 3.74 | 5.18 | 5.61 |
| **Ours** | **5.11** | **4.91** | 4.22 | **6.10** | **5.73** | **4.69** | 6.20 | 7.10 | 6.01 | 5.62 |

Table 2: Average log likelihood on SYMSOL I & II with 10% of training data.

| | 10% SYMSOL I (↑) | | | | | 10% SYMSOL II (↑) | | | |
| --- | --- | --- | --- | --- | --- | --- | --- | --- | --- |
| | *avg* | *cone* | *cyl* | *tet* | *cube* | *ico* | *avg* | *sphX* | *cylO* | *tetX* |
| Murphy et al. [14] | -7.94 | -1.51 | -2.92 | -6.90 | -10.04 | -18.34 | -0.73 | -2.51 | 2.02 | -1.70 |
| Klee et al. [12] | 2.98 | 3.51 | 2.88 | **3.62** | 2.94 | **1.94** | **3.61** | **3.12** | **3.87** | 3.84 |
| **Ours** | **3.01** | **3.63** | **3.01** | 3.53 | **3.02** | 1.91 | 3.54 | 2.88 | 3.71 | **4.04** |

**PASCAL3D+ Results** Our method achieves state-of-the-art performance on PASCAL3D+ with an average median rotation error of 9.2 degrees, as reported in Table 3. Even though object symmetries are consistently disambiguated in the labels, modeling pose as a distribution is beneficial for noisy images where there is insufficient information to resolve the pose exactly. Because our induction layer produces representations on the Fourier basis of SO(3), it naturally allows for capturing this uncertainty as a distribution over SO(3). While both our method and [12] leverage SO(3) equivariant

layers to improve generalization, we find our method achieves higher performance. We believe our induction layer is more robust to variations in how the images are rendered or captured, which is important for PASCAL3D+, since the data is aggregated from many sources. Moreover, our method does not restrict features to the hemisphere, which could be beneficial for objects, like bikes and chairs, that do not fully self-occlude their backsides.

Table 3: Rotation prediction on PASCAL3D+. First column is the average over all categories.

| | avg | plane | bike | boat | bottle | bus | car | chair | table | mbike | sofa | train | tv |
|---|---|---|---|---|---|---|---|---|---|---|---|---|---|
| | | | | | Median rotation error in degrees ($\downarrow$) | | | | | | | | |
| Mohlin et al. [47] | 11.5 | 10.1 | 15.6 | 24.3 | 7.8 | 3.3 | 5.3 | 13.5 | 12.5 | 12.9 | 13.8 | 7.4 | 11.7 |
| Prokudin et al. [35] | 12.2 | 9.7 | 15.5 | 45.6 | **5.4** | 2.9 | **4.5** | 13.1 | 12.6 | 11.8 | 9.1 | **4.3** | 12.0 |
| Tulsiani and Malik [29] | 13.6 | 13.8 | 17.7 | 21.3 | 12.9 | 5.8 | 9.1 | 14.8 | 15.2 | 14.7 | 13.7 | 8.7 | 15.4 |
| Mahendran et al. [30] | 10.1 | **8.5** | 14.8 | 20.5 | 7.0 | 3.1 | 5.1 | 9.5 | 11.3 | 14.2 | 10.2 | 5.6 | 11.7 |
| Liao et al. [33] | 13.0 | 13.0 | 16.4 | 29.1 | 10.3 | 4.8 | 6.8 | 11.6 | 12.0 | 17.1 | 12.3 | 8.6 | 14.3 |
| Murphy et al. [14] | 10.3 | 10.8 | 12.9 | 23.4 | 8.8 | 3.4 | 5.3 | 10.0 | 7.3 | 13.6 | 9.5 | 6.4 | 12.3 |
| Klee et al. [12] | 9.8 | 9.2 | 12.7 | 21.7 | 7.4 | 3.3 | 4.9 | 9.5 | 9.3 | **11.5** | 10.5 | 7.2 | 10.6 |
| Yin et al. [36] | 9.4 | 8.6 | **11.7** | 21.8 | 6.9 | **2.8** | 4.8 | **7.9** | 9.1 | 12.2 | **8.1** | 6.9 | 11.6 |
| **Ours (ResNet-50)** | 10.2 | 9.2 | 13.1 | 30.6 | 6.7 | 3.1 | 4.8 | 8.7 | 5.4 | 11.6 | 11.0 | 5.8 | 10.6 |
| **Ours** | **9.2** | 9.3 | 12.6 | **17.0** | 8.0 | 3.0 | **4.5** | 9.4 | **6.7** | 11.9 | 12.1 | 6.9 | **9.9** |

## 7    Conclusion

In conclusion, we have argued that any network that learns a three-dimensional model of the world from two-dimensional images must satisfy certain consistency properties. We have shown how these consistency properties translate into an $SO(2)$-equivariance constraint. Using the induced representation we have derived an explicit form for any neural networks that satisfies said consistency constraint. We have proposed an *induction/restriction layer*, which is a learnable network layer that satisfies the derived consistency equation. We have shown that the induction layer satisfies both a completeness property and universal property and, up to isomorphism, is unique. Furthermore, we have shown that the methods of [12, 10, 11] can be realized as specific instances of the induction layer.

The framework that we have developed is general and can be applied to other computer vision problems with different symmetries. For example, as was noted in [49], the cryogenic electronic microscopy orientation estimation problem has a latent $SO(3)$ symmetry but a manifest $SO(2) \times \mathbb{Z}_2 \cong O(2)$ (as opposed to an $SO(2)$) symmetry. With a slight modification H, the results presented in the main text allow for the construction of an induction layer that leverages this observation.

**Future Work**    In many structure-from-motion tasks, one has access to multiple images of the same object, taken at either known or unknown vantage points. Our work considers only single view pose estimation. A natural generalization of our work is to include stereo measurements into the induced/restricted representation framework. [50, 51] use transformer architectures to learn models of three-dimensional objects from two-dimensional images. Furthermore, in this work we have only considered supervised learning, but our framework naturally generalizes to unsupervised settings like [23]. Another natural extension of our work would be to include transformers into the framework presented here, which only applies to convolutional networks.

In deep learning, we often wish to construct a neural network that respects a latent symmetry $G$ that does not have an explicit action on the input data space. We have show how the induced representation can be used to construct latent $G$-equivariant neural networks. Our work provides a systematic way to construct neural architectures that accept any format of inputs and respect the latent symmetries of the problem.

## Acknowledgments and Disclosure of Funding

Owen Howell thanks the National Science Foundation Graduate Research Fellowship Program (NSF-GRFP) for financial support. Ondrej Biza, David Klee and Robin Walters were supported by National Science Foundation (NSF) grants 2107256 and 2134178. Linfeng Zhao was additionally supported by NSF grant 2107256. Ondrej Biza also acknowledges support from NSF grants 1750649

and 1763878. Owen Howell thanks Dr. Thomas Sayre-Maccord for invaluable advice and logistics help. Owen Howell further thanks Liam Pavlovic and Dr. David Rosen for useful discussions.

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
