# OpenReview forum: "Equivariant Single View Pose Prediction Via Induced and Restriction Representations"
_NeurIPS.cc/2023/Conference — NeurIPS 2023 poster_

### Official Review · Reviewer_hj9M · 2023-06-18

**Soundness:** 2 fair
**Presentation:** 2 fair
**Contribution:** 2 fair
**Rating:** 4
**Confidence:** 3

**Summary:**

This paper shows that algorithms which learn 3D representations from 2D images must satisfy certain consistency properties, which are equivalent to SO(2)-steerable constraints. To this end, a differentiable induction layer is proposed to map signals on the plane into signals on the sphere. The method is evaluated on PASCAL3D+ and SYMSOL datasets to verify the validity of performing orientation prediction.

**Strengths:**

1) A theoretical foundation for learned equivariant mappings from 2D to 3D is presented.

2) The theoretical background and analyses presented in this paper is thorough.

**Weaknesses:**

Major:
1) The implementation part is too short. It is unclear how the implementation details reflect the theoretical parts. Could the authors elaborate the connection between the theory and the implementation with more details?

2) There is no experimental analysis of how the learning-based framework satisfies the mentioned consistency and the properties. There could be some visualizations and ablation studies to show the effectiveness of the proposed method.

3) The paper mentioned that some previous methods are the special cases of the proposed learning-based method. However, from the experimental results, we can see that the proposed method does not always outperform the compared methods. In some cases, the learned layer's performance is even much worse than that of other methods. Moreover, it seems not convincing that the learned layer can avoid learning nuisance transformations. The analyses of failure cases might be helpful (for example, to explain why on some objects, the proposed method performs much worse than other methods).


Minor:
1) The size of figure 3 is too strange.

2) Many references are not complete.

3) Some references should be their conference/journal version instead of the arXiv version.

4) Some typos:

-- L35: "then"

-- L64: "a SO(2)-steerable constraints"

-- L158: "an standard"

-- L195: "a the"

-- L206: "spaces .Then"

-- L209: "respectively"

-- L230: "directions. which"

-- L300: "stero"

-- L306: "have show"

-- Figure 7: "a image"

-- Figure 7: "a induced"

-- Figure 7: "with and SO(3)"

**Questions:**

1) L33: what does "group truth" mean?

2) What is the number of layers of the default ResNet in the experiments? Why does the ResNet-50 version have very poor performance for the boat category?

**Limitations:**

From the experimental results, the learned layer can be much worse than some hand-coded methods. However, this is not well addressed in the paper. This could be mentioned as a limitation.

---

> ### Author Rebuttal · Authors · 2023-08-09
>
>
>
>
> We thank the reviewer for the thorough feedback. We address the questions posed by the reviewer below.
>
> $\textcolor{red}{ \text{Weaknesses: } }$
>
> **The implementation part is too short. It is unclear how the implementation details reflect the theoretical parts. Could the authors elaborate the connection between the theory and the implementation with more details?**
>
> Understood. This point was mentioned by multiple reviewers, we have added additional sentences on the actual implementation of our proposed method. Specifically, we have expanded line 244 in the main text:
>
> "By theorem 2 in the main text, we can expand any linear map $\Phi$ that satisfies the geometric constraint in equation 1 (line 143) as
>
> $ \\Phi (f) ( \hat{n} ) = \int_{r \in \mathbb{R}^{2} }dr \text{ } \kappa( \hat{n} , r ) f(r) $
>
> where each $\kappa$ can be written as
> $\kappa( \hat{n} , r ) = \sum_{\ell=0}^{\infty} F_{\ell}(r)^{T} Y_{\ell}(\hat{n}) $ where each $F_{\ell}(r)$ is an $SO(2)$-steerable kernels that depend on chosen input and output representations (which are user inputs). The terms $F_{\ell}(r)$ can be instantiated using the e2nn [39] package. Using the definition of $\kappa$, the decomposition of $\Phi(f)$ in terms of spherical harmonics is given by
>
> $ \\Phi ( f )( \hat{n} ) = \int_{r \in \mathbb{R}^{2} } dr \text{ } \kappa( \hat{n} , r ) f(r) = \int_{r \in \mathbb{R}^{2} } dr \text{ } [ \sum_{\ell=0}^{\infty} F_{\ell}^{T}(r) Y_{\ell}(\hat{n}) ] f(r) = \sum_{\ell=0}^{\infty} [    \int_{r \in \mathbb{R}^{2} } dr \text{ }  F_{\ell}^{T}(r) f(r)      ] Y_{\ell}(\hat{n})  $
>
> Thus, the $\ell$-th spherical harmonic coefficient of $\Phi(f)$ is given by $\Phi_{\ell}(f) = [    \int_{r \in \mathbb{R}^{2} } dr \text{ }  F_{\ell}^{T}(r) f(r)      ] $. This can be computed as a tensor contraction. The inputs to the spherical convolution are then the set of spherical harmonic coefficients $\Phi_{\ell}(f)$. Spherical convolutions are performed with the e3nn [44] package."
>
> We hope that this addendum gives the reader a better understanding of how our proposed method is implemented.
>
> **There is no experimental analysis of how the learning-based framework satisfies the mentioned consistency and the properties. There could be some visualizations and ablation studies to show the effectiveness of the proposed method.**
>
> We thank the reviewer for this excellent point. We have added one additional numerical experiment that replaces our proposed layer with a linear layer. We then train the linear layer on the SYMSOL I dataset. Post-training we measure the $SO(2)$-equivariant properties of the trained model. We compare the performance of a linear layer with our equivarient layer. We include an additional section on this numerical experiment in the attached pdf.
>
> **The paper mentioned that some previous methods are the special cases of the proposed learning-based method. However, from the experimental results, we can see that the proposed method does not always outperform the compared methods. In some cases, the learned layer's performance is even much worse than that of other methods. Moreover, it seems not convincing that the learned layer can avoid learning nuisance transformations. The analyses of failure cases might be helpful (for example, to explain why on some objects, the proposed method performs much worse than other methods).**
>
> Our method is agnostic with respect to the image formation model that the data is collected with. The exact image formation model in the ModelNet-SO(3) dataset is an orthographic projection. Thus, on the ModelNet-SO(3) dataset our model has to learn the correct image formation model while [12] already uses the correct image formation model. It should be noted that on the PASCAL3D+ dataset, where the image formation model is not described by orthographic projection, our method achieves SOTA results. By adding a residual connection to our method, we are able to achieve much better performance on the ModelNet-SO(3) dataset. We address this point in more depth in the attached pdf.
>
>
> $\textcolor{red}{ \text{Questions: } }$
>
> **L33: what does "group truth" mean?**
>
> We apologize for the typo. That should read “ground truth”. This has been changed in the text.
>
> **What is the number of layers of the default ResNet in the experiments?**
>
> We always choose the encoder to agree with existing baselines in order to create a fair comparison. On the PASCAL3D+ dataset the encoder was chosen to be ResNet-101. On the SYMSOL and ModelNet-SO(3) the encoder is a ResNet-50.
>
> **Why does the ResNet-50 version have very poor performance for the boat category?**
>
> All existing baselines use the ResNet-101 on the PASCAL3D+ dataset. The poor performance of our model with ResNet-50 encoder probably reflects the fact that the encoder is not deep enough and the boat performance is one of the most difficult categories due to pose ambiguity. We decided to include the results using the ResNet-50 encoder because they illustrate the fact that even with a weaker encoder we can still achieve competitive results.
>
> $\textcolor{red}{ \text{ Minor: } }$
> We thank the reviewer for their editing vigilance. We have addressed all of the minor issues and typos in the updated version of the paper.

---

> > ### Comment · Reviewer_hj9M · 2023-08-13
> >
> > Thanks for the authors' rebuttal.
> >
> > -- Regarding the performance boost on ModelNet-SO(3) in the rebuttal, it seems the authors need to change the network structure from the original submission, which weakens the generalizability of the original design.
> >
> > -- To better illustrate that the learning-based framework is effective, there should be some necessary visualizations as most of the previous works on equivariance did.
> >
> > -- The authors did not explain why on some objects, the proposed method performs much worse than other methods. This weakens the claim that some previous methods are the special cases of the proposed learning-based method.
> >
> > -- There are some typos in the rebuttal. Besides, it is very confusing that the caption of the last table of the attachment does not correspond to the content.

---

> > > ### Author Response · Authors · 2023-08-16
> > > **Response to Official Comment of Reviewer hj9M**
> > >
> > >
> > > We thank the reviewer for the additional comments.
> > >
> > > **Regarding the performance boost on ModelNet-SO(3) in the rebuttal, it seems the authors need to change the network structure from the original submission, which weakens the generalizability of the original design**
> > > Correct. Incorporating inductive bias into a learning algorithm is helpful, so long as the bias is correct with respect to the ground truth. Our proposed layer is agnostic with respect to the image formation model and [12] assumes that orthographic projection is the correct image formation model. On a synthetic dataset like ModelNet-SO(3), the bias of [12] accurately reflects the image formation model of the dataset. This explains why [12] outperforms our proposed layer on ModelNet-SO(3), which is a synthetic dataset. However, on a dataset of real images like PASCAL3D+, the orthographic projection assumption is incorrect and using a model with no bias (i.e. ours) yields better results. However, when our model includes the additional image formation inductive bias (which, as the reviewer points out, changes the network structure), we are able to achieve better results than [12].
> > >
> > > **To better illustrate that the learning-based framework is effective, there should be some necessary visualizations as most of the previous works on equivariance did.**
> > >
> > > Point well taken. We want this paper to be written in an intuitive way that can be understood easily. In Figure 1, we included an ‘equivarience diagram’ although in hindsight this figure does not do a good job of illustrating our idea. Specifically, the key idea is that the output of the neural network can be rotated in three dimensions. We have modified this figure.
> > > We have tried to make an additional diagram which is more illustrative. We make an additional ‘equivarience diagram’ showing both input SYMSOL images and predictions of orientations (post-training). We think this visualization is more clear than Figure 1.
> > >
> > > Lastly, at the risk of sounding slightly pompous, we think that the conditions that we derived should be better be denoted as ‘Virtual Equivarience’ or ‘Holographic Equivarience’. Specifically, the idea is that the output of the neural network (virtual model or hologram) should be rotatable in three dimensions. This requirement forces the fibers of the neural network output to transform as an $SO(3)$ representation. We think that this nomenclature may be more conceptually meaningful than “Equivariant Induced and Restricted Representations”.
> > >
> > > We could not attach figures to this comment. The additional visualizations are available at: https://anonymous-visual.github.io.
> > >
> > > **The authors did not explain why on some objects, the proposed method performs much worse than other methods. This weakens the claim that some previous methods are the special cases of the proposed learning-based method.**
> > > As explained in the main rebuttal, [12] assumes that the correct image formation model is an orthographic projection, which is the true image formation model used in the data generation of the ModelNet-SO(3) dataset. Furthermore, we claim that some previous methods are the special cases of the proposed learning-based method because they can be realized as a special case of our architecture. There is no guarantee (and nowhere do we claim) that our architecture will achieve monotonic improvement over [12] or [14]. We design the most general architecture that respects the desired symmetry constraints of the problem. We then observe experimentally that said architecture outperforms existing methods.
> > >
> > > **There are some typos in the rebuttal. Besides, it is very confusing that the caption of the last table of the attachment does not correspond to the content.**
> > > Following the request of reviewer Nd1u, we added an additional ablation study that compared the layers proposed in section E and section F.  We reran the experiments on the PASCAL3D+ dataset and compared both layers. This is shown in table 13 in the attached pdf response. We have changed the caption of table 13 to:
> > >
> > > “Comparison of $S^{2}$ and $SO(3)$ induction/restriction for Rotation prediction on PASCAL3D+. First column is the average over all categories. The feature encoder is either ResNet-50 or ResNet-101 head.”
> > >
> > > We hope that this improves the clarity of presentation.

---

### Official Review · Reviewer_Uxrh · 2023-07-06

**Soundness:** 3 good
**Presentation:** 3 good
**Contribution:** 3 good
**Rating:** 6
**Confidence:** 3

**Summary:**

This is a theory paper. The paper presents the general form of H equivariant linear maps that lift the signals defined on R2 to signals defined on S2, which is defined by convolution with kernel constraints. The main paper focuses on the case where H=SO(2) and G=SO(3). Under such a setting, the authors evaluate an SO(3) distribution regression task and show better performance there.

**Strengths:**

- The paper proposes a theoretical framework to study the group equivariance under projection, which is important when studying 3D reconstruction-related tasks
- The formulation for the SO(2)-SO(3) setting is clean and easy to implement with existing equivariant network libraries.
- The theory can be found somehow useful for the SO(3) distribution regression task.

**Weaknesses:**

- One main limitation of the proposed theory is that the authors only extensively demo the case of SO(2)—SO(3), but lack more general application settings on other groups, for example, at least SE(2) and SE(3), which has wider application settings for vision. Line 134 has an equation for the semi-direct product SE(2) but is not further illustrated as far as the reviewer understands.
- Another inevitable drawback of studying the single view SO(2)—SO(3) case is that only rotation along the camera optical axis strictly obeys the equivariance since when the object is rotated in 3D or the camera is rotated towards another direction (viewpoint) the content of the image will change due to the nature of the projection, under such case, the single view has no way to guarantee any equivariance. The reviewer thinks studying the multi-view case is more reasonable to address the view content change due to the 3D rotation. This limitation is mentioned by the author and the reviewer thinks it’s relatively fair.
- For equivariance theory paper, making it more rigorous, people always try to include the case of higher-order tensor features not just with trivial (zero-order) features. In Line 246, as far as the reviewer understands, the paper implementation only tests with zero order features, the reviewer is curious about the performance of including higher-order features.

**Questions:**

see weakness

**Limitations:**

The reviewer hasn't found explicit social impact claim.

---

> ### Author Rebuttal · Authors · 2023-08-09
>
> $\textcolor{red}{\text{Weaknesses}}$
>
> **One main limitation of the proposed theory is that the authors only extensively demo the case of SO(2)—SO(3), but lack more general application settings on other groups, for example, at least SE(2) and SE(3), which has wider application settings for vision. Line 134 has an equation for the semi-direct product SE(2) but is not further illustrated as far as the reviewer understands.**
>
> In this work, we choose to focus on pose prediction tasks only. The semi-direct product SE(2) appears in line 134, because the method that we implement is invariant with respect to in-plane translations. We have added additional sentences to the paper to make this more clear.
>
>
> **Another inevitable drawback of studying the single view SO(2)—SO(3) case is that only rotation along the camera optical axis strictly obeys the equivariance since when the object is rotated in 3D or the camera is rotated towards another direction (viewpoint) the content of the image will change due to the nature of the projection, under such case, the single view has no way to guarantee any equivariance. The reviewer thinks studying the multi-view case is more reasonable to address the view content change due to the 3D rotation. This limitation is mentioned by the author and the reviewer thinks it’s relatively fair.**
>
>  In the multi-view case, there is a more intricate set of equivarience constraints that must be satisfied. However, in this work, we chose only to consider single view pose-estimation. Although we acknowledge that much of current research focuses on multi-view problems, single view pose-estimation is an interesting problem in its own right and has applications to robotics and autonomous driving (cite works that consider single view).
>
> **For equivariance theory paper, making it more rigorous, people always try to include the case of higher-order tensor features not just with trivial (zero-order) features. In Line 246, as far as the reviewer understands, the paper implementation only tests with zero order features, the reviewer is curious about the performance of including higher-order features.**
>
> Apologies, this should read 'spherical' not 'trivial'. The 'spherical' representation of $SO(3)$ consists of one copy of each $SO(3)$ irreducible. This choice of irreducibles was also used in [12]. We fixed this typo in the text.

---

> > ### Comment · Reviewer_Uxrh · 2023-08-19
> >
> > The authors answer my main questions, however, from a practical perspective of studying 3D vision, the single view pose equivariance application is a little limited, I hope the author will include more clear clarification of the current application limitation and highlight the multiview case. Although the application has limitations, i still believe the theory part is worth for a publication, I lean to keep my score.

---

> > > ### Author Response · Authors · 2023-08-21
> > > **Response to comment of reviewer Uxrh**
> > >
> > > We acknowledge the point of the reviewer. However, we would like to emphasize to the reviewer that single view pose estimation is an interesting and applicable problem in itself. For example, single view pose estimation is an integral task in autonomous driving ( https://ieeexplore.ieee.org/document/6248074 , https://arxiv.org/pdf/2211.11962.pdf), where one needs to build models of the car environment from a single vantage point. Furthermore, pose prediction is an important task in robotic grasping (https://arxiv.org/abs/2202.03631, https://arxiv.org/abs/1809.10790 ). These problems are inherently single-view. Another problem where single view estimation is important is cryoEM, where the goal is to disentangle intrinsic molecular degrees of freedom from unknown orientational degrees of freedom (https://pubmed.ncbi.nlm.nih.gov/33542510/). For this reason, we would prefer to keep the focus on the single view case.

---

### Official Review · Reviewer_beVj · 2023-07-07

**Soundness:** 3 good
**Presentation:** 2 fair
**Contribution:** 3 good
**Rating:** 6
**Confidence:** 4

**Summary:**

This work tackles the challenging problem of achieving SO(3) equivariance over 2D projected images for pose estimation tasks. They propose a unified 2-step framework by adding an induced layer to turn SO(2)-equivariant representations into spherical signal, then using SO(3)-equivariant convolution to get the final pose distribution prediction. They have laid a general mathematical derivation of SO(3) equivariance with restriction and induction representation that satisfies the universal property, which can well summarize existing works as special cases. Comparable results have been achieved in 3 (one from supp.) well-adopted public benchmark for pose estimation.

**Strengths:**

- A novel and universal theory has been proposed to achieve SO(3) equivariance over the challenging 2D image-based pose estimation, which covers several previous works as special cases
- The proposed framework is able to extend to 6D pose estimation and monocular density reconstruction problem, which impact is potentially to be significant for a lot of other 2D-to-3D learning tasks
- Compared to the main paper, the additional full derivation in the supp. is clear and detailed

**Weaknesses:**

- My major concern is on the contribution significance, and the author might want to give a better clarification on the original contribution of the paper.
  - The claimed major contribution of induced layer seems to mainly come from this [Image to Sphere(Klee et al.)](https://arxiv.org/abs/2302.13926) paper.
  - Given we already have a lot of works tackling SO(2)-equivariant convolutions on image space, and SE(3)-equivariant operation on 3D space, it seems a trivial contribution to get a SO(3)-equivariant framework for 2D images by simply connecting existing tools from e2nn and e3nn package. Like one possible simple solution might be first lifting the 2D image using a pre-trained monocular depth, and then use SE(3) equivariant convolutions to do 6D pose estimation on the 3D space, thus we do not need to do this complex spherical signal lifting and follow-up neural operations.
  - The theory on equivariant 6D pose estimation and monocular density volume seems interesting, but it is put in the supp. without any experimental support.

- Though we have this nice equivariant property, the results are just comparable to previous SOTA methods, and on ModelNet40 it is obviously worse than the Klee et al. results, which doesn't seem to match with the conclusion drawn from the main paper.

- Sec 4.1 seems to be poorly written, and the logic flow is very confusing to understand the connections between the 3 paragraphs, like the part mapping F to S2 is missing, and I couldn't find which one is the exact induced representation.

- The theory on universal property in Sec. 5 should be put after Sec. 3, so that we can better understand the unique purpose of having both H-equivariant restriction representation and H-to-G induction representation. Also all designs in Sec. 4 can be better supported in a theoretical way before listing them out. Also the completeness property should be put in the main paper instead of only in the supp

- Line 114 should be $R^D \to S^2$, since we are mapping feature from a plane onto a sphere

**Questions:**

1. It seems that S2-conv and SO(3) are both spherical convolutions according to Sec. 6.2, I am wondering what is the major difference between these two operations.

2. The author mentions that the implicit function seems to have unique advantage in pose distribution estimation, can the propose framework be potentially extended to support implicit function prediction in the near future?

---

> ### Author Rebuttal · Authors · 2023-08-09
>
>
> We thank the reviewer for an especially detailed and thorough review. We provide a point-by-point response to the comments and questions below:
>
> $ \textcolor{red}{ \text{ Weaknesses: }} $
>
> **The claimed major contribution of induced layer seems to mainly come from this Image to Sphere(Klee et al.) paper**
>
> We would respectfully disagree with this statement. Although much of our work is heavily inspired by [12], the constraint that we formulate is a geometric statement that should be applicable to architectures that learn consistent three-dimensional models of the world from two-dimensional images. This is a statement that is geometric in nature, and is not unique to [12].
>
> **Given we already have a lot of works tackling SO(2)-equivariant convolutions on image space, and SE(3)-equivariant operation on 3D space, it seems a trivial contribution to get a SO(3)-equivariant framework for 2D images by simply connecting existing tools from e2nn and e3nn package. Like one possible simple solution might be first lifting the 2D image using a pre-trained monocular depth, and then use SE(3) equivariant convolutions to do 6D pose estimation on the 3D space, thus we do not need to do this complex spherical signal lifting and follow-up neural operations.**
>
>
> We would emphatically disagree with the reviewer on this point. There is an active area of research that aims to learn three-dimensional models of images. The problem of ‘stitching’ SO(2)-equivariant convolutions and SO(3)-equivariant convolutions is not trivial. We would ask the reviewer to find a previous paper which observes that the SO(2) $\subseteq$ SO(3) subgroup need to ‘align’, which naturally means that the map between two dimensional and three dimensional features must be an intertwiner of restricted/induced representations.
>
> In addition, the ‘lifting’ procedure proposed by the reviewer can be dependent on the camera model. For example, for some computer vision tasks, such as cryoEM, the image formation model involves Radon transformations, which is drastically different from the pinhole camera model that is applicable to telephoto-lenses. Our proposed construction allows for the camera model to be learned, instead of assumed. The only constraint imposed is geometric consistency.
>
> **The theory on equivariant 6D pose estimation and monocular density volume seems interesting, but it is put in the supp. without any experimental support.**
>
> We did not consider 6D-pose estimation in detail because the benchmarks that we consider only measure orientation estimation. We agree that this is a natural continuation of our work and we believe that our work on orientation estimation is a promising first step in this direction.
>
> **Though we have this nice equivariant property, the results are just comparable to previous SOTA methods, and on ModelNet40 it is obviously worse than the Klee et al. results, which doesn't seem to match with the conclusion drawn from the main paper.**
>
> The performance of our model on PASCAL3D+ and SYMSOL I is current state of the art. Using a residual connection, we can achieve better results on the ModelNet-SO(3) dataset. We address this point in more detail the attached pdf.
>
>
> **Sec 4.1 seems to be poorly written, and the logic flow is very confusing to understand the connections between the 3 paragraphs, like the part mapping F to S2 is missing, and I couldn't find which one is the exact induced representation**
>
> We have added to section 4.1 and section E. Please see response to reviewer mVf3.
>
> **The theory on universal property in Sec. 5 should be put after Sec. 3, so that we can better understand the unique purpose of having both H-equivariant restriction representation and H-to-G induction representation. Also all designs in Sec. 4 can be better supported in a theoretical way before listing them out. Also the completeness property should be put in the main paper instead of only in the supp**
>
> We cannot fit the full derivation of the completeness property in the main text. We will include a statement of the completeness property in the main text and then refer the reader to the proof in the appendix.
>
> **Line 114 should be R2 -> S2, since we are mapping feature from a plane onto a sphere**
>
> We are interested in mapping the space of functions $f: \mathbb{R}^{2} \rightarrow \mathbb{R}^{d}$ into the space of functions $ g : S^{2} \rightarrow \mathbb{R}^{d'} $. We have changed line 114 to
>
> "For convenience we ignore discritization and treat the feature maps as having continuous inputs $f : \mathbb{R}^{2} \rightarrow \mathbb{R}^{d}$. To leverage spatial symmetries in 3D, we would like to map features $f$ into the space of features defined on a sphere: $g : S^{2} \rightarrow \mathbb{R}^{d'}$. We thus need to consider the space of maps between two function spaces.
>
> $ \textcolor{red}{ \text{Questions: }} $
>
> **It seems that S2-conv and SO(3) are both spherical convolutions according to Sec. 6.2, I am wondering what is the major difference between these two operations.**
>
> Spherical convolutions and SO(3) convolutions are distinct operations. Spherical convolution (cite taco paper) is an operation that convolves two signals defined on $S^{2}$ (cite taco paper) and returns is a signal defined on SO(3). SO(3) convolution is an operation that takes as inputs two signals defined on the group SO(3) and the output of an SO(3) convolution is a signal defined on SO(3). In the appendix, we include an ablation study that compares mapping directly to the sphere and performing spherical convolutions vs. mapping directly to SO(3).
>
> **The author mentions that the implicit function seems to have unique advantage in pose distribution estimation, can the propose framework be potentially extended to support implicit function prediction in the near future?**
> Yes it is possible to generalize our work to implicit function prediction. This observation was raised by a few of the reviewers and we address this point in the attached pdf.

---

> > ### Author Response · Authors · 2023-08-16
> > **Referenced Citation**
> >
> > The referenced citation should be:
> > 1. Spherical convolution: https://arxiv.org/pdf/1801.10130.pdf

---

> > ### Comment · Reviewer_beVj · 2023-08-20
> >
> > Thanks for the authors' rebuttal, and my concerns on the significance and writing are moderately addressed. The updated results on ModelNet-40 look interesting to me, does that mean general improvement on the architecture actually matters more to improve performance?
> >
> > Given the potential impact of such a work on several downstream tasks, I am also wondering the efficiency of the proposed method, does the performance boost by introducing different equivariant convolutions also sacrifice a lot on the memory and inference speed compared to Klee et al?

---

> > > ### Author Response · Authors · 2023-08-20
> > > **Response to comment of reviewer beVj**
> > >
> > > We thank the reviewer for additional comments on our manuscript.
> > >
> > > We are a bit unsure what the sentence: **does that mean general improvement on the architecture actually matters more to improve performance?** means. Can the reviewer please rephrase the question?
> > > If the reviewer is asking if there are ways to combine existing neural network methods with the geometric constraint derived in our text; the answer is most likely yes. We would expect that combining more advanced vision architectures such as feature pyramids networks (c.f. https://arxiv.org/pdf/1612.03144.pdf) or vision based transformer methods (https://arxiv.org/abs/2103.14030) with induced/restricted equivarience constraints would lead to improved performance, although this is outside the scope of this work.
> > >
> > > **Given the potential impact of such a work on several downstream tasks,**
> > > We are pleased that the reviewer recognizes the importance of designing equivariant neural networks for monocular vision tasks.
> > >
> > >
> > > **I am also wondering the efficiency of the proposed method, does the performance boost by introducing different equivariant convolutions also sacrifice a lot on the memory and inference speed compared to Klee et al?**
> > >
> > > Our proposed method incurs slight overhead relative to Klee [5] in terms of memory and inference speed. Our proposed layer is implemented efficiently by a matrix multiplication and a pooling operation, which can be done in roughly the same time as the orthogonal projection in [5].
> > >
> > > Naively, one would think that making the projection of [5] learnable adds many additional free parameters. However, equivalence constraints drastically limit the number of allowed parameters. Specifically, for the PASCAL3D+ architecture [5] has roughly 42.663M
> > >  trainable parameters (including backbone), and our proposed method has roughly 52.222M
> > >  traininable parameters (including backbone). This is a ~%20 percentage increase in the number of trainable parameters.

---

> > > > ### Comment · Reviewer_beVj · 2023-08-20
> > > >
> > > > Thanks for the clarification, my concerns have been well addressed.

---

### Official Review · Reviewer_mVf3 · 2023-07-08

**Soundness:** 3 good
**Presentation:** 2 fair
**Contribution:** 3 good
**Rating:** 6
**Confidence:** 2

**Summary:**

The paper studies learning equivariant SO3 representations from 2D images. The authors first propose a generalized theory that shows SO2 equivariance constraint for image to spherical signals. Then they propose a construction algorithm called induction layer to implement equivariance. The proposed method outperforms relevant baselines on pose estimation on SYMBOL and PASCAL 3D+.

**Strengths:**

The paper studies an important problem in 3D vision  (equivariant-SO3 from 2D input) and shows its empirical advantage in 3D pose estimation. This has a potential impact on lots of downstream tasks.
They provide a unified theory that shows previous works are their special cases.
The empirical results are convincing, especially results with 10% training data.


**Weaknesses:**

1. It is not clear how to construct the induction layer from the paper. More details may be provided on how to implement the layer, e.g. how to solve equation 2.
2. The experiments are object-centric 6D pose estimation. I wonder if the method can generalizes to cluttered scenes like those in BOP challenges (https://bop.felk.cvut.cz/challenges/)



**Questions:**

Please see weakness section

**Limitations:**

The limitation is discussed.

---

> ### Author Rebuttal · Authors · 2023-08-09
>
>
> Thank you for the feedback. Regarding your questions about our work, here is our response:
>
> $\textcolor{red}{ \text{Weaknesses: }} $
>
> **It is not clear how to construct the induction layer from the paper. More details may be provided on how to implement the layer, e.g. how to solve equation 2**
>
> We have added an additional details on the form of the solution of the geometric constraint and how this is implemented in PyTorch. We also would like to mention that if reading code is more helpful than reading mathematics, our code is attached in the original submission. We have tried to make the code as user friendly as possible.
>
> We have expanded line 244 in the main text:
>
> "The filters in the induction layer were instantiated using the e2nn [39] package."
>
> By theorem 2 in the main text, we can expand any linear map $\Phi$ that satisfies the geometric constraint in equation 1 (line 143) as
>
> $ \\Phi (f) ( \hat{n} ) = \int_{r \in \mathbb{R}^{2} }dr \text{ } \kappa( \hat{n} , r ) f(r) $
>
> where each $\kappa$ can be written as
> $\kappa( \hat{n} , r ) = \sum_{\ell=0}^{\infty} F_{\ell}(r)^{T} Y_{\ell}(\hat{n}) $ where each $F_{\ell}(r)$ is an $SO(2)$-steerable kernels that depend on chosen input and output representations (which are user inputs). The terms $F_{\ell}(r)$ can be instantiated using the e2nn [39] package. Using the definition of $\kappa$, the decomposition of $\Phi(f)$ in terms of spherical harmonics is given by
>
> $ \\Phi ( f )( \hat{n} ) = \int_{r \in \mathbb{R}^{2} } dr \text{ } \kappa( \hat{n} , r ) f(r) = \int_{r \in \mathbb{R}^{2} } dr \text{ } [ \sum_{\ell=0}^{\infty} F_{\ell}^{T}(r) Y_{\ell}(\hat{n}) ] f(r) = \sum_{\ell=0}^{\infty} [    \int_{r \in \mathbb{R}^{2} } dr \text{ }  F_{\ell}^{T}(r) f(r)      ] Y_{\ell}(\hat{n})  $
>
> Thus, the $\ell$-th spherical harmonic coefficient of $\Phi(f)$ is given by $\Phi_{\ell}(f) = [    \int_{r \in \mathbb{R}^{2} } dr \text{ }  F_{\ell}^{T}(r) f(r)      ] $. This can be computed as a tensor contraction. The inputs to the spherical convolution are then the set of $\Phi_{\ell}(f)$. Spherical convolutions are performed with the e3nn [44] package.
>
>
> **The experiments are object-centric 6D pose estimation. I wonder if the method can generalizes to cluttered scenes like those in BOP challenges**
>
> We thank the reviewer for bringing this benchmark to our attention. In its current form, our method would be unable to deal with pose-estimation for multiple objects. Most equivarient methods have a similar problem as they are equivarient with respect to global transformations. Specifically, there is not a natural way to rotate a single object in a scene and not rotate the whole scene. One natural possible method to address pose-estimation in cluttered scenes would be to first segment objects then apply pose-estimation to each object. This is an interesting problem and will be left for future work.

---

### Official Review · Reviewer_Nd1u · 2023-07-30

**Soundness:** 2 fair
**Presentation:** 2 fair
**Contribution:** 3 good
**Rating:** 5
**Confidence:** 1

**Summary:**

The paper focuses on the task of 3d pose prediction. The paper proposes a set of consistency constraints for learning 3d representation and shows that few of the previously proposed neural architectures follow these constraints. Further using their constraints they propose a new architecture with an induction layer that maps feature maps to a feature sphere. Using this new architecture they show that their method can achieve state-of-the-art performance on PASCAL3D+ and Symsol 2 dataset.

**Strengths:**

-> the paper proposes a unifying theory that encompases previously proposed architecture

-> the paper achieves strong performance on PASCAL3D+ and SYMSOL II

-> the paper is well written and described

**Weaknesses:**

-> I couldn't find any ablations/analysis in the paper

-> No results on ModelNet which has been included in previous papers

-> The differences between baseline in terms of architecture/pre-training/losses used has not been made very clear

**Questions:**

-> How does the method compare on datasets such as ModelNet or CO3D? I would be interested in seeing comparisions against recent methods like these: https://arxiv.org/abs/2208.05963

->  Do baselines use the same backbone/pre-training?

-> What happens if u drop the induction layer while still modelling the uncertainity?

-> How does increasing/decreasing the training data affect the results? (currently only 10%)

-> Why results on SYMSOL II are poor compared to the baseline? Anything specific about SYMSOL II?

**Limitations:**

Yes

---

> ### Author Rebuttal · Authors · 2023-08-09
>
> Thank you very much for the useful feedback. Please find our point-by-point response below.
>
> $\textcolor{red}{ \text{ Weaknesses: }} $
>
>
>  **I couldn't find any ablations/analysis in the paper**
>
> We have added two ablations studies. The first study compares the performance an architecture that uses a spherical convolution followed by an SO(3) convolution and an architecture that utilizes just SO(3) convolutions. We have also added one additional ablation study that compares a naive linear layer vs our proposed layer. We compare both the performance and equivarience error of the linear layer and our proposed layer. These results are included in the attached pdf.
>
> **No results on ModelNet which has been included in previous papers**
>
> We have included results on ModelNet-SO(3) in the appendix. Other reviewers pointed out that the performance of our architecture was not SOTA on ModelNet-SO(3). Using a residual connection, we can achieve better results on the ModelNet-SO(3) dataset. We address this point in more detail the attached pdf.
>
>
> **The differences between baseline in terms of architecture/pre-training/losses used has not been made very clear**
>
> In order to ensure that the reader understands our work, we have added additional clarifications to the main paper.
> We used the same backbone (ResNet-50 for SYMSOL and ModelNet, ResNet-101 for PASCAL3D+) and pre-training (ImageNet-1K) as the baselines. The loss was trained using cross-entropy on SO(3), which was also used in [14].
>
>
> $ \textcolor{red}{ \text{ Questions: }} $
>
>
> **How does the method compare on datasets such as ModelNet or CO3D?**
>
> As mentioned before, we have included the results on ModelNet in the appendix. These results are also commented on in the attached pdf. We do not have time to evaluate our method on the CO3D dataset, but we agree this would be an interesting experiment.
>
> **Do baselines use the same backbone/pre-training?**
>
> Yes. In order to make a fair comparison, we used the same backbone (ResNet-50 for SYMSOL and ModelNet, ResNet-101 for PASCAL3D+) and pre-training (ImageNet-1K) as the baselines. Interestingly, we are able to achieve very competitive results on PASCAL3D+ even when using the ResNet-50 backbone. This is somewhat surprising as the ResNet-50 backbone does not have the same representational power as the ResNet-101.
>
> **What happens if u drop the induction layer while still modeling the uncertainty?**
>
> We are a bit unsure about what the reviewer means by this statement. The uncertainty is important to modeling objects with intrinsic symmetries. Many of the existing baselines use models which model uncertainty in some way, see for example [14].  In the attached pdf we replaced the induction/restriction layer with a linear layer and compared performance. Does this address the stated question?
>
>
>
> **How does increasing/decreasing the training data affect the results? (currently only 10 percent)**
>
> In general, equivariant methods perform better than non-equivariant methods in the low-data regime. This is because equivariant methods can generalize to unseen data if said data is related to previously seen data by symmetry transformation. In the regime of very large amounts of data, equivariant and non-equivariant methods tend to have similar performance [4].
>
>
> **Why results on SYMSOL II are poor compared to the baseline? Anything specific about SYMSOL II?**
>
> This question was asked by multiple reviewers. Unlike SYMSOL I where a single model is trained on all class, in SYMSOL II a separate model is trained on each distinct class. We address this point further attached pdf.

---

### Author Rebuttal · Authors · 2023-08-10

We thank the reviewers for their comments on our manuscript. We are pleased that the majority of reviewers recognize the fact that our theory provides a unifying perspective for a set of relevant computer vision tasks:
- Reviewer Nd1u: "the paper proposes a unifying theory that encompases previously proposed architecture"
- Reviewer mVf3: "The paper studies an important problem in 3D vision (equivariant-SO3 from 2D input) and shows its empirical advantage in 3D pose estimation. This has a potential impact on lots of downstream tasks."
- Reviewer beVj: "A novel and universal theory has been proposed to achieve SO(3) equivariance over the challenging 2D image-based pose estimation, which covers several previous works as special cases"
- Reviewer Uxrh: "The paper proposes a theoretical framework to study the group equivariance under projection, which is important when studying 3D reconstruction-related tasks"

We recognize the criticism of the reviewers in regards to clarity of presentation, numerical performance on benchmarks and dearth of ablation studies. In order to address these concerns, we have added an additional description of the implementation (found in section 4.2) of our proposed method. In the attached pdf, we include tables for additional numerical experiments. Specifically,

- We include an additional experiment on ModelNet-SO(3) dataset [33]. A major concern of many of the reviewers was that the performance of our architecture was worse that [12] on the ModelNet-SO(3). In some ways, this may be expected as the [12] assumes that the correct image formation model is an orthographic projection, which is the true image formation model used in the data generation of the ModelNet-SO(3) dataset. Our architecture needs to learn the correct image formation model. By including additional biases about the image formation model, we can achieve state of the art results on the ModelNet-SO(3) dataset. We added a residual connection to our induction/restriction layer that is an orthographic projection. This reflects the assumption that for the ModelNet-SO(3) model, the true image formation model is close to orthographic projection, which is common for pinhole camera models [2]. With this additional bias, our model achieves SOTA when averaged over each ModelNet-SO(3) category. These results are shown in the attached pdf in table 10.

- We include an ablation study where we replace our construction with a linear layer. We replaced the induction/restriction layer with a linear layer and trained on the SYMSOL I dataset [14]. We chose the SYMSOL dataset as it consists of rotated solids and any model that performs well on SYMSOL should be approximately equivarient. We choose the spherical layer to have fibers transforming in the $ \\rho _{spherical} = \bigoplus _{\ell=0}^{6} D^{\ell} $ SO(3)-representation. Post-training, we then tested the SO(2)-equivarience properties of the output spherical layer we found a percentage error of about %18 with
output SO(2)-representation approximately $\text{Res} _{SO(2)}^{SO(3)}( \\rho _{spherical}  ) $. This simple numerical experiment shows that the trained linear layer approximately satisfies the geometric constraint derived in the main text. These results are shown in table 12.

- A concern of the reviewers was the performance of our architecture on the SYMSOL II dataset. This phenomena is expected and also observed in [12]. Unlike SYMSOL I on the SYMSOL II dataset, a different model is trained on each class independently. Thus, the SYMSOL I task is more concerned with learning while the SYMSOL II is more concerted with representational power. The method proposed in [14] has significantly more parameters than both our method and [12]. Generalization of our proposed framework to implicit models is an interesting direction for future work. These results are shown in table 11.

- We include an ablation study which compares $S^{2}$ and $SO(3)$ convolutions. We rerun the experiments in the main text using an induction layer that maps images directly to SO(3). The direct induction to SO(3) slightly underperforms the induction to $S^{2}$ on the PASCAL3D+ dataset. We believe that this adds some depth to section E and section F.  These results are shown in table 13.

We addressed specific comments directly in the response section.

Minor:
There is a mistake in the appendix in section J.4. Specifically, the constraints imposed by restricted/induced representations are conditions on the intertwiners between layers, not the content of irriducibles in each layer. Line 836- 845 should be deleted and replaced with: "Specifically, the linear map between boundary layers must satisfy,

$ \\Psi \in \text{Hom}_{H}(  ( \\rho _{i} , \mathcal{F} _{i}^{H}  ) ,   \text{Res} _{H}^{G}(  \sigma _{1} , \mathcal{F} _{1}^{G}  )   ) \cong \text{Hom} _{G}(   \text{Ind} _{H}^{G}( \\rho _{i} , \mathcal{F} _{i}^{H}  ) , (\sigma _{1} , \mathcal{F} _{1}^{G} )    )$

Specifically, if
$( \rho _{i} , \mathcal{F} _{I}^{H} )$
and
$( \sigma _{1} , \mathcal{F} _{1}^{G} ) $
decompose into irreducibles as

$ ( \rho _{i} ,	\mathcal{F} _{i}^{H} ) = \bigoplus _{ \rho \in \hat{H} } m _{i \rho } (\rho , V _{\rho} ) , \quad \quad ( \sigma _{1} , \mathcal{F} _{G}^{1} ) = \bigoplus _{ \sigma  \in \hat{G} } n _{1 \sigma } ( \sigma  , W _{\sigma}  )  $


Then, we can write the induced and restricted representations in terms of the branching and induction rules,

$ \text{Res} _{H}^{G}( ( \sigma _{1} , \mathcal{F} _{1}^{1})  ) = \bigoplus _{ \rho \in \hat{H} }( (\sum _{  \sigma \in \hat{G} } n _{1 \sigma } B _{ \sigma, \rho} ) ( \rho , V _{\rho} ) ) \quad \text{Ind} _{H}^{G}( ( \rho _{i} , \mathcal{F} _{i}^{H} )    ) = \bigoplus _{ \sigma \in \hat{G} }(  ( \sum _{  \rho \in \hat{H} } m _{ i, \rho } I _{\rho, \sigma}) ( \sigma , W _{\sigma} ) ) $

and intertwiners $\Psi$ can be computed by considering the decomposition of irreducibles."

---

### Decision · Program_Chairs · 2023-09-21

**Decision:**

Accept (poster)

**Comment:**

The paper proposes a method for single view single object 3D pose estimation by mapping feature maps to feature spheres. The reviewers raised concerns regarding lack of ablations, lack of experiments in ModelNet, as well as decreased performance w.r.t specific baselines. The rebuttal addressed those concerns by providing experiments in ModelNet, ablation studies and explaining that the baseline knows the image formation model, which their method doesn’t and needs to infer it. Overall, the reviewers appreciate the idea of the paper and encourage publication.